# Gender inequalities of psychosomatic complaints at work vary by occupational groups of white- and blue-collar and level of skill: A cross sectional study

Julia Grasshoff *, Batoul Safieddine, Stefanie Sperlich, Johannes Beller

Department for Medical Sociology, Hannover Medical School, Hanover, Germany

* Grasshoff.Julia@mh-hannover.de

**Data Availability Statement:** This paper uses data from the BIBB/BAuA Employment Survey of the Working Population on Qualification and Working Conditions in Germany 2018. The survey was

## Abstract

### Background

Previous research has shown that women report more psychosomatic complaints at work than men. However, knowledge about gender inequalities in psychosomatic complaints within occupational groups and specific symptoms is lacking. This study aims to compare gender inequalities in psychosomatic complaints in the occupational groups of white-collar high-skilled, white-collar low-skilled, blue-collar high-skilled and blue-collar low-skilled workers.

### Methods

The study implemented a cross sectional design using data from the nationwide German Employment Survey of the Working Population on Qualification and Working Conditions conducted in 2017/ 2018. Psychosomatic complaints were operationalised by the following symptoms: headache, insomnia, tiredness, irritability, dejection, physical fatigue, and emotional fatigue. N = 20012 working German-speaking respondents were sampled. After excluding persons with missing data on the study variables, the sample consisted of N = 16359 persons.

### Results

Women reported significantly more psychosomatic complaints than men in the subgroups of white-collar high-skilled and white-collar low-skilled ($ps < .05$), inequalities in blue-collar high-skilled and blue-collar low-skilled only being numerical. Regarding specific symptoms, women reported more psychosomatic complaints then men in the subgroups of white-collar high-skilled workers, white-collar low-skilled workers, and blue-collar low-skilled workers. Headaches, physical fatigue, and emotional fatigue were the most common symptoms. The white-collar high-skilled subgroup had the highest number of symptoms with significant gender inequalities. These effects remained after controlling for age, working hours, parental status and marital status.

conducted by the Federal Institute for Vocational Education and Training (BIBB), and the Federal Institute for Occupational Safety and Health (BAuA). The data access was provided via a Scientific-Use-File of the Data Research Centre at the Federal Institute for Vocational Training and Education (BIBB-FDZ). The data are available on request (https://www.bibb.de/de/140.php) for scientific purposes after having signed an agreement with the owner.

**Funding:** Open Access funding enabled and organized by Projekt DEAL. Funded by a grant from the Ministry for Science and Culture of Lower Saxony awarded to Dr. Johannes Beller for his project "Modern Work - Healthy Work? Change in Work-Related Physical Activity as an Explanatory Factor in Physical and Psychological Morbidity Development" (Funded of ressources of SPRUNG).

**Competing interests:** The authors have declared that no competing interests exist.

## Conclusions

Gender inequalities in psychosomatic complaints are ubiquitous but vary in their frequency by occupational subgroup and specific psychosomatic complaint. Women in white-collar high-skilled jobs in particular report to be burdened more often by many specific psychosomatic symptoms. Future studies should investigate the reasons for these occupational inequalities and develop interventions to reduce health inequalities in the workplace.

## Introduction

### Gender-related psychosomatic inequalities at work

Extensive evidence exists on occupational gender inequalities. Previous research shows that women report more psychosomatic complaints at work than men as physical and emotional fatigue, sadness, irritability and muscle, heart, and stomach pain [1–5]. Papers used different styles of operationalisation as yes/no questions and sum scores of questionnaires, but the general trend was found again and again. Women visit the doctor more often and have more frequent and more severe complaints, especially non-specific symptoms such as headaches and fatigue [3, 6]. Women call in sick at work more often [3], although men and women do not seem to differ in their norms and attitudes regarding the appropriateness of calling in sick above a certain level of psychosomatic complaints and illness [7]. Psychosomatic symptoms are an important predictor and reason for work absence and work disability [8, 9].

### Gender-related occupational inequalities at work

Women and men tend to have different occupations, occupy different hierarchical positions at work, have different work demands and tend to work less hours [3, 5, 10]. In Germany, gender segregation within occupations has decreased only slightly since the 1970s [10–12]. Women also take on more care work alongside their paid work than men [13]. In order to differentiate the effect of gender on the burden, it is necessary to consider different occupational subgroups.

### Occupation-related psychosomatic inequalities at work

Research on inequalities in psychosomatic complaints between occupational subgroups is rare and shows conflicting results. Myrtik et al. [14] found that white-collar workers showed higher levels of psychosomatic complaints, while Axelrod & Gavin [15] and Schreuder et al. [16] found more psychosomatic complaints among blue-collar workers. However, level of qualification was not taken into account in those studies. Regarding stresses, in blue-collar jobs, workers are more exposed to physical strains as chemicals and heavy weights [3, 17], while white-collar jobs rather lead to psychosocial and emotional demands [3]. In addition to work intensity, emotional demands are the most important predictor to impaired occupational health resulting in burnout [18]. It is yet to be researched if these stresses differ in impairment on different genders. Overall, for all genders, conditions as longer and asocial working hours (especially more than fulltime), emotional demands, demands for hiding emotions, high demands and low control, discrimination, low job promotion, low predictability, low meaning of work low social support have been found to have a strong negative effect on psychosomatic complaints when comparing workers with more or less of these stresses with each other [3, 17, 19–23]. Hünefeld and Dötsch [5] found that all workers in female-dominated jobs report more psychosomatic complaints then workers in male-dominated or equally distributed

working sectors. Additionally, research on gender inequalities in occupational subgroups in psychosomatic complaints are lacking.

### How are occupational and gender inequalities related?

Many gender inequalities in work have been established, but it remains to be investigated to which degree gender inequalities in psychosomatic complaints occur within occupational subgroups. The current study aimed to help filling this gap in the literature, by investigating gender inequalities in psychosomatic complaints within occupational subgroups. In addition, the role of working hours, family and parental status will be considered, and specific psychosomatic complaints were investigated. We ask: "Does the level of psychosomatic complaints differ between genders in occupational subgroups?" and "Are these inequalities stable over all psychosomatic symptoms?". Based on the literature it is to be expected that women are more burdened regarding psychosomatic complaints in all occupational subgroups and symptoms. The research questions could be answered in the following contributing differentiated insights into the topic.

## Methods

A cross-sectional study design was implemented. Ethics approval was not required as this study was not experimental but used data that has already been collected and published. We used data from the nationwide German Employment Survey of the Working Population on Qualification and Working Conditions conducted from October 2nd 2017 to April 5th 2018 [24, 25]. The survey was implemented by the Federal Institute for Vocational Education and Training (BIBB) in cooperation with the Federal Institute for Occupational Safety and Health (BAuA). All participants in the survey provided informed consent to participate.

Participants were recruited randomly from a national telephone registry. The sample includes German-speaking employees in Germany aged 15 and over who were engaged in paid employment for at least ten hours per week. Individuals who had interrupted their occupation for a maximum period of three months (e.g., parental leave) were included. Individuals were interviewed in 1:1 telephone interviews using structured questionnaires. In order to include the growing number of persons who do not have a landline, 30% of the participants were contacted by mobile phone. Data was accessed for research purposes by the main author on 2nd of May 2023. Authors did not have access to information that could identify individual participants. Data access was provided as a Scientific-Use-File (SUF) of the Data Research Centre at the Federal Institute for Vocational Training and Education (BIBB-FDZ). As the data are owned by this third-party, the authors are not the data owners, but only users. The dataset is available for academic purposes. All underlying data are freely available in a public repository on request after having signed an agreement with the owner (https://www.bibb.de/de/1403.php). SUFs are distributed directly via the BIBB-FDZ. For this purpose, an application form (available online via https://www.bibb.de/dokumente/pdf/BIBB_FDZ_Antrag_SUF_FDZ_deutsch.pdf) must be completed, signed and sent to the BIBB-FDZ by post (BIBB—Bundesinstitut für Berufsbildung Arbeitsbereich 1.5: Forschungsdatenzentrum, Postfach 20 12 64, 53142 Bonn, Germany) or e-mail (fdz@bibb.de). Data access and permission to use for research has been granted by the BIBB-FDZ to the researchers Johannes Beller and Julia Graßhoff.

### Psychosomatic and demographic measures

Our main variables of interest were seven psychosomatic complaints in the workplace by asking the participants "Please tell me if the following health complaints occurred to you during

the past 12 months while you were at work or on workdays. We are interested in the complaints that occurred frequently." Participants could choose to respond with "yes" or "no" regarding the following symptoms: headache, insomnia, tiredness, irritability, dejection, physical fatigue, and emotional fatigue. To operationalise an overall psychosomatic complaint score, the responses of the strains were treated as binary variables which were summed to an overall score. The score was treated as a quantitative variable in the analyses. As covariates we included gender, age, occupation, hours of work per week, marital status, and parental status. Gender was treated as a binary variable ("male"/ "female") in the analyses. Age was treated as a metric variable. Regarding occupation, participants could self-select their job title. This was recoded into the variable "collar" via the International Standard Classification of Occupations (ISCO88), which classifies jobs regarding their skill level (describing the complexity and range of tasks of an occupation with focus on the tasks that require the highest skill level) and their skill specialisation (describing the field of work, required tools and what kind of goods or services are produced) [26]. As the nature of skills acquisition differs in the extent of formal training and experience, classification is based on the type of skills required to perform the job and how they are needed instead of their acquisition. The sample was divided into the occupational groups "white-collar high-skilled" (including legislators, senior officials, managers, professionals, technicians and associate professionals), "white-collar low-skilled" (including clerks, service workers, shop and market sales workers), "blue-collar high-skilled" (including skilled agricultural and fishery workers and craft and related trades workers) and "blue-collar low-skilled" (including plant and machine operators, assemblers and elementary occupations). Occupational group was treated as an ordinal variable in the analyses. Regarding hours of work per week participants could answer any number. This value was divided into intervals: "up to 20 hours", "21–25 hours", "26–30 hours", "31–35 hours", "36–40 hours", "more than 40 hours. The variable was treated as an ordinal variable in the analyses. Gross income was included as continues variable for demographic data. Regarding marital status, participants could choose to respond with "married", "registered partnership", "divorced", "widowed" or "single". The options "married", and "registered partnership" were combined to the option "married". The variable was treated as a categorical variable in the analyses. Parental status was treated as a binary variable ("yes"/ "no") in the analyses, both genetic and adoptive children were included in the category "yes". Overall, N = 20012 were sampled. After excluding persons with missing data on study variables listwise (18% of the original sample), a final sample size of N = 16359 resulted.

## Data analysis

After checking the descriptive data, analyses were performed using chi2-tests, linear and logistic regressions. To screen for multicollinearity, a correlation matrix was calculated with all independent variables. To test for inequalities between genders regarding the total score of psychosomatic complaints, linear regression was applied by for each subgroup. To test for inequalities in the proportions of specific psychosomatic symptoms, Chi2-tests of independence were applied. To calculate adjusted effect sizes, in the linear regression analyses the total score of psychosomatic complaints and in logistic regression the individual burdens were used as dependent variables. As a large number of tests (7 symptoms x 4 occupational subgroups = 28 comparisons) were performed, a stricter $\alpha$-level than the usual 5% was set for the Chi2-tests and regressions analysis. The Bonferroni-Correction was applied. It takes the increasing probability of type-1- error into account by adjusting the $\alpha$-level to the number of tests performed. Hence, the corresponding formula is $\alpha_{corrected} = \alpha_{chosen\ a\ priori} / n$. As the a priori $\alpha$ was 0.05 and 28 comparisons where tested, the final formula was $\alpha_{corrected} = 0.05 / 28 = 0.002$. We used

cross-sectional weights which are assumed to produce a nationally representative sample. The weighting variable "gew2018" was provided by BAuA, details regarding its determination can be found in the BAuA method article [25]. StataMP 15 software was used for all analyses.

# Results

## Basic characteristics of the study sample

As depicted in Table 1, 50.96% of participants were male. Male participants were on average 46.20 years old ($SD$ = 0.13), female participants were on average 47.50 ($SD$ = 0.12) years old. Of all male participants, 60.56% worked in white-collar high-skilled jobs, 9.54% in white-collar low-skilled jobs, 16.57% in blue-collar low-skilled jobs and 13.34% in blue-collar low-skilled jobs. Of all female participants, 67.28% worked in white-collar high-skilled jobs, 24.40% in while-collar low-skilled jobs, 2.84% in blue-collar high-skilled jobs and 13.34% in blue-collar low-skilled jobs. Of all male participants, white-collar high-skilled workers earned on overage

**Table 1. Descriptive statistics of the sample.**

|  | Male | Female |
|---|---|---|
| Sample size N/ % | 8336 (50.96%) | 8023 (49.04%) |
| Age in years, mean (SD) | 46.20 (0.13) | 47.50 (0.12) |
| Occupational groups, %, |  |  |
| White-collar high-skilled | 60.56% | 67.28% |
| White-collar low-skilled | 9.54% | 24.40% |
| Blue-collar high-skilled | 16.57% | 2.84% |
| Blue-collar low-skilled | 13.34% | 5.47% |
| Gross income in Euro, mean (SD) |  |  |
| White-collar high-skilled | 5077.27 (63.95) | 5077.27 44.81) |
| White-collar low-skilled | 3218.20 (119.38) | 5077.27 29.42) |
| Blue-collar high-skilled | 3057.40 (54.24) | 2048.90 (74.85) |
| Blue-collar low-skilled | 2687.05 (56.94) | 1436.57 (45.70) |
| Marital status groups, % |  |  |
| Married | 9.54% | 24.40% |
| Single | 16.57% | 2.84 |
| Divorced | 13.34% | 5.47 |
| Widowed | 1.31% | 4.10% |
| Parental status, % |  |  |
| Having children | 61.79% | 73.23 |
| Not having children | 38.21% | 26.77% |
| Working hours per week groups, % |  |  |
| up to 20 hours | 4.93% | 17.23% |
| 21–25 hours | 1.46% | 8.84% |
| 26–30 hours | 3.23% | 13.35% |
| 31–35 hours | 4.23% | 10.37% |
| 36–40 hours | 37.14% | 28.29% |
| 41–45 hours | 23.87% | 12.68% |
| More than 44 hours | 25.13% | 9.25% |

To screen for multicollinearity, a correlation table was calculated for age, marital status, parental status and working hours. All correlations were between -0.37 and 0.20. Therefore, no problems with multicollinearity are to be expected.

5077.27 euros ($SD$ = 63.52), white-collar low-skilled earned 3218.20 ($SD$ = 119.38) euros, blue-collar high-skilled workers earned 3057.40 (SD = 54.24) euros, blue-collar low-skilled workers earned 2687.05 ($SD$ = 56.94) euros. Of all female participants, white-collar high-skilled workers earned on overage 3258.07 euros ($SD$ = 44.81), white-collar low-skilled earned 1939.05 ($SD$ = 29.42) euros, blue-collar high-skilled workers earned 2049.90 (SD = 74.86) euros, blue-collar low-skilled workers earned 1436.57 ($SD$ = 45.70) euros. Of all male participants, 55.05% were married, 33.90% were single, 9.74% were divorced, 1.31% were widowed and 61.79% had children. Of all female participants, 55.62% were married, 24.40% were single, 15.58% were divorced, 4.10% were widowed and 73.23% had children. Of all male participants, the majority with 37.14% worked 35 to 40 hours per week. Of all female participants, the majority with 28.29% worked 35 to 40 hours per week. For detailed information, see Table 1.

## Occupation-dependent psychosomatic gender inequalities

Looking at the psychosomatic complaint score by gender as depicted in Table 2, an overall mean difference of $Diff_{male/female}$ = 0.9 points emerged between the two groups, meaning women being more burdened by almost every symptom. The mean difference was significant ($p < .001$, $d = 0.25$). Taking a closer look, women were more affected in all occupational sub-groups. The mean difference in white-collar high-skilled jobs $Diff_{male/female}$ = 0.68 points ($p < .001$, $d = 0.31$), white-collar low-skilled jobs $Diff_{male/female}$ = 0.22 points ($p = .039$, $d = 0.1$) were significant. The mean differences in blue-collar high-skilled jobs $Diff_{male/female}$ = 0.1 points ($p = .507$, $d = 0.18$) and in blue-collar low-skilled jobs $Diff_{male/female}$ = 0.38 points ($p = .283$, $d = 0.05$) were not significant. Next, the effects were tested via linear regression analyses controlling for age, parental status, marital status and working hours. As depicted in Table 2, the adjusted $p$ values remained consistent with a suppression effect occurring in the blue-collar low-skilled group turning into a significant mean difference ($p = .003$).

Additionally, the regression analysis showed that over all occupational groups, the adjusted $p$ value for the influence of working hours on psychosomatic complaint were significant ($p < .001$) with people working more reporting more complaints.

Next, Chi2 tests were used to investigate stratified mean differences of the occupation subgroups per burden using the Bonferroni-corrected α-levels (see Table 3). The relationship between gender and burden was significant in all burdens in the white-collar high-skilled group, women being more burdened ($p < .001$). In the white-collar low-skilled group, women were significantly more burdened in headaches ($p = .001$). In the blue-collar high-skilled group, no significant mean differences were found. In the blue-collar low-skilled group, women were significantly more burdened in emotional fatigue ($p = .001$). Logistic regression analyses were used to study whether gender still predicted the overall psychosomatic complaint score when controlling for age, parental status, marital status and working hours. In the white-collar high-skilled group, the adjusted $p$-values remained consistent. Headaches had the strongest effect size: Women had 2.08 times to the odds of reporting headaches as compared to

**Table 2. Gender differences in the psychosomatic complaint score, stratified by occupational groups.**

| Group | Male Mean / SD | Female Mean /SD | P | Adjusted P | Adjusted effect size |
|---|---|---|---|---|---|
| All | 1.96 / 2.09 | 2.50 / 2.28 | $< .001$ | $< .001$ | 0.25 |
| White-collar high-skilled | 1.88 / 2.08 | 2.56 / 2.3 | $< .001$ | $< .001$ | 0.31 |
| White-collar low-skilled | 2.23 / 2.23 | 2.45 / 2.25 | .039 | $< .001$ | 0.10 |
| Blue-collar high-skilled | 2.02 / 2.04 | 2.12 / 2.22 | .507 | .121 | 0.05 |
| Blue-collar low-skilled | 2.01 / 2.09 | 2.39 / 2.2 | .283 | .003 | 0.18 |

Note. Adjusted effect size = standartised regression coefficient

**Table 3. Symptomatic mean differences in occupational groups.**

| Variable | Male „Yes"in % | Female „Yes"in % | P | Adjusted P | Adjusted effect size |
|---|---|---|---|---|---|
| **Subgroup: White-collar high-skilled** | | | | | |
| Headache | 28.65 | 42.57 | < .001 | < .001 | 2.08 |
| Insomnia | 26.68 | 35.03 | < .001 | < .001 | 1.43 |
| Tiredness | 43.07 | 52.87 | < .001 | < .001 | 1.61 |
| Irritability | 24.70 | 31.62 | < .001 | < .001 | 1.40 |
| Dejection | 15.35 | 21.62 | < .001 | < .001 | 1.55 |
| Physical fatigue | 24.58 | 36.57 | < .001 | < .001 | 1.72 |
| Emotional fatigue | 25.44 | 35.40 | < .001 | < .001 | 1.71 |
| **Subgroup: White-collar low-skilled** | | | | | |
| | Male „Yes"in % | Female „Yes"in % | P | Adjusted P | Adjusted effect size |
| Headache | 31.45 | 40.14 | < .001 | < .001 | 1.75 |
| Insomnia | 30.31 | 31.92 | .432 | .203 | 1.19 |
| Tiredness | 49.94 | 49.44 | .266 | .013 | 1.38 |
| Irritability | 30.19 | 29.53 | .541 | .227 | 1.18 |
| Dejection | 20.88 | 23.29 | .100 | .008 | 1.49 |
| Physical fatigue | 34.34 | 38.00 | .138 | .006 | 1.43 |
| Emotional fatigue | 25.53 | 29.93 | .022 | .001 | 1.58 |
| **Subgroup: Blue-collar high-skilled** | | | | | |
| | Male „Yes"in % | Female „Yes"in % | P | Adjusted P | Adjusted effect size |
| Headache | 23.24 | 28.51 | .059 | .004 | 1.49 |
| Insomnia | 25.27 | 23.27 | .941 | .856 | 0.92 |
| Tiredness | 46.63 | 43.86 | .360 | .936 | 0.99 |
| Irritability | 25.27 | 25.88 | .353 | .253 | 1.11 |
| Dejection | 20.35 | 23.25 | .564 | .514 | 1.21 |
| Physical fatigue | 43.16 | 45.18 | .297 | .229 | 1.12 |
| Emotional fatigue | 17.89 | 21.93 | .145 | .200 | 1.34 |
| **Subgroup: Blue-collar low-skilled** | | | | | |
| | Male „Yes"in % | Female „Yes"in % | P | Adjusted P | Adjusted effect size |
| Headache | 24.28 | 32.80 | .175 | .001 | 1.88 |
| Insomnia | 29.59 | 33.26 | .732 | .043 | 1.44 |
| Tiredness | 45.68 | 50.80 | .775 | .094 | 1.33 |
| Irritability | 24.19 | 25.51 | .989 | .299 | 1.22 |
| Dejection | 20.86 | 24.60 | .886 | .518 | 1.14 |
| Physical fatigue | 38.94 | 47.84 | .402 | .068 | 1.37 |
| Emotional fatigue | 17.90 | 24.37 | .017 | .001 | 1.89 |

Note: Adjusted effect size = standartised regression coefficient / odds ratio

men. In the white-collar low-skilled group, a suppression effect appeared: the adjusted *p*-values regarding mean differences in emotional fatigue changed from not significant to significant. All other *p*-values in this group remained consistent. Headaches had the strongest effect size: Women had 1.75 times the odds of reporting headaches as compared to men. In the blue-collar high-skilled-group, all *p*-values in this group remained consistent. Again, headaches had the strongest effect size: Women had 1.49 times the odds to report headaches as compared to men. In the blue-collar low-skilled group, the suppression effect appeared with headaches and emotional fatigue. All other *p*-values in this group remained consistent. Emotional fatigue had the strongest effect size: Women had 1.89 times the odds to report it as compared to men.

## Discussion

### Key results

We investigated how gender inequalities in psychosomatic complaints vary across occupational subgroups and regarding specific psychosomatic complaints. We analysed a German sample of more than 16.000 working people aged 15 and over. Considering the large sample size, the low correlations between the independent variables and using population weights, the results of the analyses are likely to be generalisable for the whole working population. We found that women were more burdened in most analysed subgroups of occupation, both at the total score of psychosomatic complaints and at specific symptoms. These effects were most pronounced in the occupational subgroup of white-collar high-skilled workers.

The result that women generally having more psychosomatic complaints at work is in line with previous findings [1–5]. A new finding of the current study is, that these gender inequalities vary across occupational subgroups. They were found in significant extent in the occupational subgroups of white-collar high-skilled workers and white-collar low-skilled workers. This effect also remained stable when controlling for age, working hours, parental status and marital status. In terms of effects sizes, gender inequalities in psychosomatic symptoms were most severe in high-skilled white-collar jobs. Regarding gender inequalities in specific symptoms, in the white-collar high-skilled group, women were significantly more burdened in all symptoms (headaches, insomnia, tiredness, irritability, dejection, physical fatigue and emotional fatigue). Effect sizes showed that women had 1.71 to 2.08 times the odds of having them compared to men. In the white-collar low-skilled group, women were significantly more burdened by headaches and emotional fatigue. In the blue-collar high-skilled group, there were no significant mean differences. In the blue-collar low-skilled group, women were significantly more burdened by emotional fatigue. It is noticeable, that women working in white-collar high-skilled jobs tend to have a considerably higher risk of experiencing a high number of symptoms than their male colleagues, which was to be expected after analysing the total score. Therefore, the current study adds to the literature that the specific occupational groups differ how gender inequalities in psychosomatic complaints occur.

Additionally, the regression analysis showed psychosomatic complaints increased with rising working hours per week, which is in line with previous literature [17]. A new insight is, that this can be found even when controlling for gender, parental, marital status and especially collar.

### Relevance for the population

The effect sizes in this study are medium-sized. However, a medium effect at a population level still affects many people. A medium change in the mean can lead to a disproportionate change at the tails of the distribution as simulated in Carey et al. [27]. When setting a threshold for a certain level of burden (e.g., when people decide to call in sick at work), the larger the population size, the more people will cross that threshold. In the case of psychosomatic complaints, in a clinical context a medium effect or a symptom further up may not have much impact on the life of a person with moderate distress. However, when looking at a whole population, this means that if the mean score of complaints is higher in women, a large number of women will move into the higher parts of the distribution, perhaps being distressed enough to call in sick at work or even meeting the criteria of a depressive episode, as the measured symptoms are all frequent symptoms of a depressive episode [28]. Further research is needed to pinpoint the threshold of symptoms that are "enough" for people to call in sick. Although a higher number of symptoms may represent a mild or moderate depressive episode, people differ in their ability to cope, adapt, and function in everyday life. However, having these symptoms

increases the likelihood of taking time off work [29]. Interestingly, regarding gender inequalities in the connection of symptoms and sick leave, it was found that men had a stronger association between depressive symptoms and sick leave, although the heterogeneity of the study outcomes were far wider for men than for women [29]. This stands in contrast to findings of women calling in sick more often. However, those numbers do not differentiate between reasons for sick leave, so other symptoms might play a role for women and when a far higher number of women feel too burdened to work then men, those numbers can still be consistent.

## Explanations

As correlates, the results cannot explain why women are more burdened. One possibility is that men and women appear to work different tasks, despite having the same job title [5, 30]. Interestingly, the occupational subgroup of white-collar high-skilled workers not only has the most pronounced gender inequalities in complaints but is also the only subgroup with an almost 50/50 gender distribution in the sample. So, equality in employment doesn't automatically lead to equality in burden. This is in line with Hünefeld and Dötsch's [5] findings, that more burdened women are found in male-dominated, female-dominated, and equally distributed jobs. Although the research shows contradictions, in most studies, the assessment of factors making a job environment more or less stressful do not seem to differ much between genders [3, 4, 19], women may be exposed to those more often. It was found that women are more exposed to tasks and stresses that negatively influence well-being as being exposed to other people's suffering, simultaneous tasks, being interrupted [31], emotional demands, demands for hiding emotions, discrimination, [18] and having less control and freedom planning their amount of work [2, 32]. At the same time, women report more freedom to plan and organize their set work, higher job security and having more social support through colleagues and superiors which improves psychological and physical health [2, 30, 33]. Also, stresses differ between blue- and white-collar jobs as more physical stresses in blue-collar jobs [3, 17] and psychosocial and emotional stresses in white-collar [3]. It remains unclear if and why these inequalities in stresses and their connection of inequalities in psychosomatic complaints appear more in white-collar high-skilled jobs than in other occupational subgroups.

There might be certain stresses, tasks and resources that predominantly influence psychosomatic complaints that women are more exposed to or react to more. Hünefeld and Dötsch [5] found, that psychosomatic complaints are especially high in female-dominated jobs. There might be a main effect of gender and work field. Looking more closely at working conditions in the occupational subgroups and whether gender inequalities remain when these are controlled for, we can get an idea of which working conditions exactly are factors in the inequalities in burden. In order to reduce health inequalities, it is important to investigate, why women are more burdened and identify changes in their working conditions.

As another possibility, reporting of complaints is subjective. Although subjective health status is sufficiently correlated with objective health outcomes [34], it could itself be influenced by gender. Nevertheless, gender differences in pain sensitivity, as described by Pieretti et al. [35], do not seem to explain the overall inequalities, as occupational effects occurred. If gender inequalities were purely explained by differences in reporting we would expect similar results across occupational groups; this was not the case in our study, strongly suggesting additional reasons. Although men and women do not seem to differ in their norms and attitudes regarding the appropriateness of calling in sick above a certain level of psychosomatic complaints and illness of a described worker [7], judging others compassionately does not automatically mean judging oneself in the same way. Therefore, although subjective reporting might have influenced the observed gender inequalities, it does not fully explain them.

As a last possibility, women and men also strongly differ in their non-working lifestyle, with women, for example, being much more burdened by informal care activities [13]. However, in our study, when controlling for age, working hours, parental status and marital status, significant results remained significant and certain non-significant trends became significant, especially in the blue-collar. This suppression effect indicates that certain demographic characteristics might play an additional role in the gender inequalities in psychosomatic complaints, which should be investigated further.

## Limitations

Regarding the validity and reliability of the questionnaire, the items are not part of clinically validated questionnaires as the Becks-Depression-Inventory or the Center for Epidemiological Studies Depression Scale. Especially the self-reporting of specific symptoms may differ in their validity, as a symptom as headaches may be easier to self-assess then irritability, which needs a higher level of reflection capability. Also, there is no differentiation between trivial and severe symptoms. An interview with an experienced clinician and a clinically validated questionnaire enables more valid and reliable data. Regarding generalisability, it should be kept in mind that this study, being a survey study, might suffer from selectivity and non-response bias [36, 37]. The survey tried to minimise the problem of non-response. That included conducting the sample with a mobile phone share of 30%, the use of experienced, well-trained, and appropriately remunerated interviewers, the definition of a sufficiently long field time and fixed specifications for the scheduling and number of contact attempts. It must be kept in mind, that people with high psychosomatic stress may participate less often in surveys; a correlation between poor subjective health and non-responding was found when comparing groups of responders and non-responders in follow-up-surveys [38, 39]. Also, employees in good health at work are more likely to stay in the labour market at all, qualifying themselves for the survey our data comes from. As especially the highest part of the distribution of extent of burden is important when it comes to sick-leave and disability, this self-selection may let the sample seem healthier than the population is and reduce statistical effects.

## Opportunities for future research

As to future research, it seems important to investigate trends over time. Gender inequalities in psychosomatic complaints may have changed and possibly even increased over recent years and may increase in the future. Modern work offers more white-collar high-skilled jobs [10] and the number of people graduating from high school and university before entering the workforce is growing [40, 41], thus potentially increasing overall work-related gender inequalities in psychosomatic health. Thus, relatively more people in this occupational subgroup cause bigger gender inequalities in the population as a whole. Time-trend analyses can map this and will be an important addition to better understand the evolution of gender inequalities in modern work. In addition to that, mean differences of age groups, marital and parental statuses can be stratified. For further research, a comparison between Germany, being a comparatively gender-fair work field compared to other more unequal countries in the world might be interesting as well.

## Supporting information

**S1 Data.**
(DOCX)

## Author Contributions

**Conceptualization:** Julia Grasshoff, Batoul Safieddine, Stefanie Sperlich, Johannes Beller.

**Data curation:** Julia Grasshoff.

**Formal analysis:** Julia Grasshoff.

**Funding acquisition:** Johannes Beller.

**Methodology:** Julia Grasshoff, Johannes Beller.

**Software:** Julia Grasshoff.

**Supervision:** Johannes Beller.

**Writing – original draft:** Julia Grasshoff.

**Writing – review & editing:** Julia Grasshoff, Batoul Safieddine, Stefanie Sperlich, Johannes Beller.

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
