## [Decision Letter · Decision Letter 0]

9 Feb 2024

PONE-D-23-38613Gender differences of psychosomatic complaints at work vary by occupational groups of white- and blue-collar and level of skill: A cross sectional studyPLOS ONE

Dear Dr. Grasshoff,

Thank you for submitting your manuscript to PLOS ONE. After careful consideration, we feel that it has merit but does not fully meet PLOS ONE’s publication criteria as it currently stands. Therefore, we invite you to submit a revised version of the manuscript that addresses the points raised during the review process.

We look forward to receiving your revised manuscript.

Kind regards,

Rocco Franco

Academic Editor

PLOS ONE

Journal Requirements:

"Open Access funding enabled and organized by Projekt DEAL. Funded by a grant from the Ministry for Science and Culture of Lower Saxony awarded to Dr. Johannes Beller for his project “Modern Work - Healthy Work? Change in Work-Related Physical Activity as an Explanatory Factor in Physical and Psychological Morbidity Development” (Funded of ressurces of SPRUNG)."

3. In the online submission form, you indicated that "This paper uses data from the BIBB/BAuA Employment Survey of the Working Population on Qualification and Working Conditions in Germany 2018. The survey was conducted by the Federal Institute for Vocational Education and Training (BIBB), and the Federal Institute for Occupational Safety and Health (BAuA). The data access was provided via a Scientific-Use-File of the Data Research Centre at the Federal Institute for Vocational Training and Education (BIBB-FDZ). The data are available on request (https://www.bibb.de/de/1403.php) for scientific purposes after having signed an agreement with the owner."

4. Please include a copy of Table 3 which you refer to in your text on page 10.

**Additional Editor Comments:**

Please revise this paper according Reviewer's recommendation

Regards

Reviewers' comments:

Reviewer's Responses to Questions

**Comments to the Author**

1. Is the manuscript technically sound, and do the data support the conclusions?

Reviewer #1: Yes

Reviewer #2: Partly

2. Has the statistical analysis been performed appropriately and rigorously? 

Reviewer #1: Yes

Reviewer #2: No

3. Have the authors made all data underlying the findings in their manuscript fully available?

Reviewer #1: Yes

Reviewer #2: Yes

4. Is the manuscript presented in an intelligible fashion and written in standard English?

Reviewer #1: Yes

Reviewer #2: Yes

5. Review Comments to the Author

Reviewer #1: Dear Authors,

The paper is very interesting and highlights gender inequalities that can be avoided.

Technically the paper is simple and at the same time correct.

The authors are aware of the limitations of this type of study and have controlled for them.

I would like to make a few comments.

When you do a sex-stratified analysis, you have to study the differences.

If these differences are avoidable, then we would talk about "inequalities". Then we can talk about gender instead of sex.

I think this terminology (e.g., in the title) should be better revised throughout the text.

As a small comment, in Table 3 (which puts Table 1) the Chi-squared column should be removed. This is a suggestion, as I do not think it adds anything. If the authors decide to leave this column, I have nothing to say.

Reviewer #2: This descriptive study assesses the extent of gender differences in psychosomatic complaints at work by different occupational subgroups in a study of employees in Germany. The strength of this paper is its examination of gender differences separately by occupational subgroup. However, the main weakness of the paper in its current form is the lack of discussion around the different contexts in these occupational subgroups.

My main comment is that this study needs much more engagement with previous literature to motivate the examination of these gender differences by occupational subgroup. The paper currently focuses on how they differ in terms of gender composition, but there are other important differences between these subgroups. White collar and blue collar jobs have differences in physical job demands while low-skill and high-skill jobs presumably have large differences in terms of their psychosocial demands. These job characteristics could differentially impact men and women and should be discussed in both the intro to motivate the analyses and in the discussion to interpret the findings.

Can the authors provide citation(s) for how other studies have operationalized psychosomatic complaints? And some more background about why these complaints are likely to be affected by work?

I have several questions about the treatment of several analysis variables:

How was skill dichotomized into high vs low skill?

Why were working hours categorized as “up to 10 hours worked per week”, “11-20 hours worked per week”, “21-30 hours worked per week”, and “30 or more hours worked per week”?

Very few participants work 10 or fewer hours per week so it is unclear why this needs its own category. Would it be possible to divide the over 30 hours per week into multiple categories to identify those working long hours? Previous literature examining the effects of working hours on adverse outcomes has found more than full-time work to be a risk factor, yet the current categorization does not allow the authors to examine this.

It would be clearer to refer to the “Child status” variable instead as “parental status”

The statistical analyses need some sort of correction for multiple testing. The analyses presented in Table 3 represent 28 comparisons (7 symptoms x 4 occupational subgroups), so it is not surprising that several are statistically significant.

Throughout the results section, the interpretations of the odds ratios are incorrect and overstate the findings. For example, the interpretation of an odds ratio of 2.08 as “women were 2.08 times more likely to report headaches as compared to men” should be “women had 2.08 times the odds of reporting headaches as compared to men”

6. PLOS authors have the option to publish the peer review history of their article (what does this mean?). If published, this will include your full peer review and any attached files.

Reviewer #1: **Yes: **José Fernández-Sáez

Reviewer #2: No

---

## [Author Response · Author response to Decision Letter 0]

8 Apr 2024

Comments of the Reviewers (also uploaded as a file where changes are highlighted in colour):

Reviewer 1

Comment 1 Reviewer 1: When you do a sex-stratified analysis, you have to study the differences. If these differences are avoidable, then we would talk about "inequalities". Then we can talk about gender instead of sex. I think this terminology (e.g., in the title) should be better revised throughout the text.

Answer 1 R 1: Thank you for your review of our manuscript and referring to a more precise wording. We changed the word “differences” to “inequalities” in the title and the manuscript as gender differences may be avoidable under other working conditions. We only kept the word “differences” when directly referring to numerical mean differences in the result section. 

Comment 2 Reviewer 1: As a small comment, in Table 3 (which puts Table 1) the Chi-squared column should be removed. This is a suggestion, as I do not think it adds anything. If the authors decide to leave this column, I have nothing to say.

Answer 2 R 1: Thank you for pointing out how to make the manuscript more compact. We deleted the column as we agree that it does not add additional information.

Reviewer 2

Comment 1 Reviewer 2: My main comment is that this study needs much more engagement with previous literature to motivate the examination of these gender differences by occupational subgroup. The paper currently focuses on how they differ in terms of gender composition, but there are other important differences between these subgroups. White collar and blue collar jobs have differences in physical job demands while low-skill and high-skill jobs presumably have large differences in terms of their psychosocial demands. These job characteristics could differentially impact men and women and should be discussed in both the intro to motivate the analyses and in the discussion to interpret the findings.

Answer 1 R 2: Thank you for your review of our manuscript and pointing out the chance to include a wider range of perspective on the stresses that white- and blue-collar jobs. We additionally extended the following paragraph with additional literature as Doef (1999) Artazcoz (2016) and Ganster (2018) in the introduction:

“Regarding stresses, in blue-collar jobs, workers are more exposed to physical strains as chemicals and heavy weights [3, 17], while white-collar jobs rather lead to psychosocial and emotional demands which significantly lead to exhaustion and reduced well-being [3]. In addition to work intensity, emotional demands are the strongest predictor for impaired occupational health and burnout [18]. It is yet to be researched if these stresses differ in impairment on different genders. Overall, for all genders, conditions as longer and asocial working hours (especially more than fulltime), emotional demands, demands for hiding emotions, high demands and low control, discrimination, low job promotion, low predictability, low meaning of work low social support have been found to have a strong negative effect on psychosomatic complaints when comparing workers with more or less of these stresses with each other [3, 17, 19, 20, 21, 22, 23].”

Additionally, we did so in the discussion:

“Also, stresses differ between blue- and white-collar jobs as more physical stresses in blue-collar jobs [3, 17] and psychosocial and emotional stresses in white-collar [3]. It remains unclear if and why these inequalities in stresses and their connection of inequalities in psychosomatic complaints appear more in white-collar high-skilled jobs than in other occupational subgroups. There might be certain stresses, tasks and resources that predominantly influence psychosomatic complaints that women are more exposed to or react to more.”

Comment 2 Reviewer 2: Can the authors provide citation(s) for how other studies have operationalized psychosomatic complaints? And some more background about why these complaints are likely to be affected by work?

Answer 2 R 2: Thank you for pointing out to be more precise in the citation of our literature. To get a fuller impression of recent studies, we added the following paragraph in the introduction (changes highlighted in green): 

“Previous research shows that women report more psychosomatic complaints at work than men as physical and emotional fatigue, sadness, irritability and muscle, heart, and stomach pain [1-5]. Papers used different styles of operationalisation as yes/no questions and sum scores of questionnaires, but the general trend was found again and again.”

Additionally, the paragraph for the introduction we presented in Comment 1 includes more information and literature where different working conditions were analysed in connection with complaints indicating that certain work(-ing conditions) leads to more or less complaints. 

Comment 3 Reviewer 2: How was skill dichotomized into high vs low skill?

Answer 2 R 2: The coding system of ISCO88 made job descriptions for a range of jobs and rated their skill requirements according to the work task that acquired the highest skill to do it (an additional information that was added in the manuscript as well). Occupations were grouped into several supergroups regarding skill and specialisation. To give a better insight which occupations were put where we adjusted the following paragraph in the methods section: 

“The sample divided into the occupational groups “white-collar high-skilled” (including legislators, senior officials, managers, professionals, technicians and associate professionals), “white-collar low-skilled” (including clerks, service workers, shop and market sales workers), “blue-collar high-skilled” (including skilled agricultural and fishery workers and craft and related trades workers) and “blue-collar low-skilled” (including plant and machine operators, assemblers and elementary occupations).”

Thus, skill wasn’t dichotomised per se but rated together with skill specialisation which resulted in the four occupational groups.

Comment 4 Reviewer 2: Why were working hours categorized as “up to 10 hours worked per week”, “11-20 hours worked per week”, “21-30 hours worked per week”, and “30 or more hours worked per week”? Very few participants work 10 or fewer hours per week so it is unclear why this needs its own category. Would it be possible to divide the over 30 hours per week into multiple categories to identify those working long hours? Previous literature examining the effects of working hours on adverse outcomes has found more than full-time work to be a risk factor, yet the current categorization does not allow the authors to examine this.

Answer 4 R 2: Thank you for pointing out the impact of working hours on stress. We added literature to these findings (e.g, Eurofound, 2017) and changed the subgroups into 5-hour-steps between 20 hours and 45+ hours to make the possible extra burden of working over 30 or 40 hours/ week better measurable. In the results section and discussion, we included the results of the regression analysis that our data validated the effect that longer working hours led to more complaints although the effect of collar remains when controlling for it.

Comment 5 Reviewer 2: It would be clearer to refer to the “Child status” variable instead as “parental status”.

Answer 5 R2: To clarify the wording, we changed the words in the text as suggested. 

Comment 6 Reviewer 2: The statistical analyses need some sort of correction for multiple testing. The analyses presented in Table 3 represent 28 comparisons (7 symptoms x 4 occupational subgroups), so it is not surprising that several are statistically significant.

Answer 6 R2: To improve the quality of the analysis, we added a Bonferroni correction using a stricter α-level in the analysis. Using the formular αcorrected= αchosen a priori / n with n= 7x4= 28 comparisons. This resulted in αcorrected= 0.05 / 28 = 0.002. 0.002 was used as a stricter cut-off for significance. Some values did not meet the significance anymore, the overall trend and statement that white-collar high-skilled female workers are dominantly under complaint inequalities remained. 

Comment 7 Reviewer 2: Throughout the results section, the interpretations of the odds ratios are incorrect and overstate the findings. For example, the interpretation of an odds ratio of 2.08 as “women were 2.08 times more likely to report headaches as compared to men” should be “women had 2.08 times the odds of reporting headaches as compared to men.”

Answer 7 R2: Thank you for pointing out this error of interpretation. We adjusted the wording in the result section as suggested.

---

## [Decision Letter · Decision Letter 1]

1 May 2024

Gender inequalities of psychosomatic complaints at work vary by occupational groups of white- and blue-collar and level of skill: A cross sectional study

PONE-D-23-38613R1

Dear Dr. Grasshoff,

We’re pleased to inform you that your manuscript has been judged scientifically suitable for publication and will be formally accepted for publication once it meets all outstanding technical requirements.

Kind regards,

Rocco Franco

Academic Editor

PLOS ONE

Additional Editor Comments (optional):

Dear Authors

This manuscript may be accepted and publishable in PlosOne,also based on the reviewers' recommendations,

Congratulations

Reviewers' comments:

Reviewer's Responses to Questions

**Comments to the Author**

1. If the authors have adequately addressed your comments raised in a previous round of review and you feel that this manuscript is now acceptable for publication, you may indicate that here to bypass the “Comments to the Author” section, enter your conflict of interest statement in the “Confidential to Editor” section, and submit your "Accept" recommendation.

Reviewer #1: All comments have been addressed

Reviewer #2: All comments have been addressed

2. Is the manuscript technically sound, and do the data support the conclusions?

Reviewer #1: Yes

Reviewer #2: Yes

3. Has the statistical analysis been performed appropriately and rigorously? 

Reviewer #1: Yes

Reviewer #2: Yes

4. Have the authors made all data underlying the findings in their manuscript fully available?

Reviewer #1: Yes

Reviewer #2: Yes

5. Is the manuscript presented in an intelligible fashion and written in standard English?

Reviewer #1: Yes

Reviewer #2: Yes

6. Review Comments to the Author

Reviewer #1: The authors have clarified all my comments.

I think this paper can be published in its current state.

Has improved a lot.

I have nothing else to say

Reviewer #2: (No Response)

7. PLOS authors have the option to publish the peer review history of their article (what does this mean?). If published, this will include your full peer review and any attached files.

Reviewer #1: No

Reviewer #2: No

---

## [Editor Report · Acceptance letter]

10 May 2024

PONE-D-23-38613R1 

PLOS ONE

Dear Dr. Grasshoff, 

I'm pleased to inform you that your manuscript has been deemed suitable for publication in PLOS ONE. Congratulations! Your manuscript is now being handed over to our production team.

Kind regards, 

on behalf of

Dr. Rocco Franco 

Academic Editor

PLOS ONE